# Techniques for Symbol Grounding with SATNet

**Sever Topan**[1, 2]**, David Rolnick**[1, 3, 4]**, and Xujie Si**[1, 3, 4]

[1]McGill University, [2]NVIDIA, [3]Mila – Quebec AI Institute, [4]CIFAR AI Research Chair
`{stopan, drolnick, xsi}@cs.mcgill.ca`

## Abstract

Many experts argue that the future of artificial intelligence is limited by the field's ability to integrate symbolic logical reasoning into deep learning architectures. The recently proposed differentiable MAXSAT solver, SATNet, was a breakthrough in its capacity to integrate with a traditional neural network and solve visual reasoning problems. For instance, it can learn the rules of Sudoku purely from image examples. Despite its success, SATNet was shown to succumb to a key challenge in neurosymbolic systems known as the *Symbol Grounding Problem*: the inability to map visual inputs to symbolic variables without explicit supervision ("label leakage"). In this work, we present a self-supervised pre-training pipeline that enables SATNet to overcome this limitation, thus broadening the class of problems that SATNet architectures can solve to include datasets where no intermediary labels are available at all. We demonstrate that our method allows SATNet to attain full accuracy even with a harder problem setup that prevents any label leakage. We additionally introduce a *proofreading* method that further improves the performance of SATNet architectures, beating the state-of-the-art on Visual Sudoku.

## 1   Introduction

Recent years have seen significant advancements in deep learning, providing breakthroughs in image, video, and audio processing [1]. Despite its success, deep learning has many known limitations, such as low interpretability, vulnerability to adversarial attacks, and difficulty in solving problems requiring hard logical constraints [2–5]. To overcome these limitations, experts have described the need to migrate from purely deep learning-based systems to neurosymbolic artificial intelligence systems, which integrate neural networks with logical reasoning [6]. In this work we focus on improving a promising development in this field: the award-winning architecture known as SATNet [7].

SATNet is a differentiable MAXSAT solver based on a low-rank semidefinite relaxation approach. It can be integrated into traditional Deep Neural Networks (DNNs) to solve composite learning problems that require both logical reasoning and visual understanding. One such problem is Visual Sudoku, where the model must learn the rules of a Sudoku puzzle purely from visual examples. When trained end-to-end, SATNet is able to achieve 63.2% total board accuracy in this task while traditional DNN architectures are unable to exceed 0% [7]. This was regarded as a significant breakthrough for neurosymbolic architectures. However, it was recently noted that SATNet training relies upon "leakage" of labels through the logical constraint layer to the DNN used to classify digits [8].

This leakage essentially means that SATNet is learning in two supervised stages, where it first trains its digit classification component under direct supervision, and only then trains its SAT layer to learn the logical constraints delineating Sudoku. When the leakage is removed, SATNet's ability to solve Visual Sudoku drops to 0% [8]. This is significant, because taken independently, these two sub-problems are significantly easier. Digit classification is considered a solved problem, and while SAT constraint mining is more difficult, it could be argued that the differentiable aspect is no longer beneficial if the system needs supervision on its inputs to learn regardless. For instance, there exist other SAT constraint miners that are not differentiable but outperform SATNet [9]. Overall, the issue

35th Conference on Neural Information Processing Systems (NeurIPS 2021).

of being unable to learn to solve composite visual reasoning problems end-to-end is referred to as the *Symbol Grounding Problem*, and is considered one of the fundamental prerequisites for artificial intelligence to perform practical logical reasoning [8, 10].

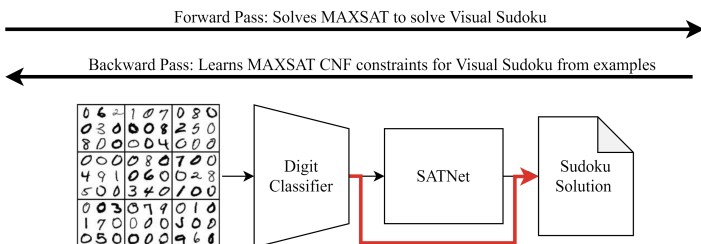

Figure 1: The SATNet architecture used to solve Visual Sudoku. The red line shows the label leakage issue, which when removed, results in the Symbol Grounding Problem.

We observe a key challenge of symbol grounding is the large gap between the compositional nature of logical reasoning and the end-to-end gradient-based nature of neural networks. The former helps to reduce a sophisticated reasoning system into simple, independent modules, each of which can be designed manually or learned, while the latter encourages fusing all components together and using gradients as a universal means for learning. Many recent approaches aim to bridge this gap by relaxing logical constraint solving through numerical optimisations [11–14]. Although such end-to-end gradient-based optimisation is appealing, it can fail to address seemingly simple tasks like Visual Sudoku. The success of SATNet is in fact due to *inadvertent* supervision of intermediate modules. We argue that compositionality does not have to be the opposite of the end-to-end learning design. The latter is particularly preferable because it eliminates the need for supervision of intermediate modules, which is often required by a compositional design. If compositionality can be trained using self-supervision (i.e. without manual effort), compositionality would then be at least equally preferable. This is the approach that we take in the present work, synergistically combining compositionality with end-to-end learning without any explicit intermediate supervision. We envision our methodology forming a new paradigm for tackling neurosymbolic learning.

We describe a self-supervised pre-training method that can be used to bootstrap SATNet in order to overcome the Symbol Grounding Problem. Our methodology enables us to tackle a class of what we call *Ungrounded* MAXSAT problems, where label data are available only for the output variables of the MAXSAT problem. In the Visual Sudoku case, this formulation manifests itself as a dataset where, as before, inputs consist of images of digits describing the input cells of a Sudoku board. The labels of the dataset, however, consist of numerical representations only for the board cells that were not given as inputs. This means that there is no way of identifying what digit each input image refers to except by learning the rules of the Sudoku puzzle *in parallel* to predict the non-input values. We refer to this problem as *Ungrounded Visual Sudoku*. We show that our method improves the state of the art on this problem from 0% to 64.8%, achieving similar performance on Ungrounded Visual Sudoku as SATNet with label leakage does in the grounded version of the same problem. In short, our main contributions are the following:

1. We describe a self-supervised clustering and distillation process for training a visual classifier within a SATNet architecture.

2. We introduce a *Symbol Grounding Loss* that makes it possible to train logical constraint layers on an ungrounded symbol representation.

3. We show empirically that our methodology allows SATNet to achieve full performance on *ungrounded* Visual Sudoku (where label leakage is impossible), a task where previous state-of-the-art was 0%.

4. We introduce a *Proofreader* that improves the performance of any SATNet system (grounded or ungrounded), achieving state-of-the-art performance on Visual Sudoku.

## 2 Background

Our contribution draws from several areas. We begin with preliminaries describing the problem, before discussing related work.

### 2.1 The Problem

MAXSAT, the optimisation analog of SAT, represents a rich set of problems to which many program complexity classes can be reduced. A MAXSAT Solver $\mathcal{S}$ aims to maximally satisfy a set of $n$ boolean clauses over $m$ variables by modulating the values of the variables. These clauses are typically written in Conjunctive Normal Form, and represented numerically as a matrix $M \in \mathbb{R}^{n \times m}$. We can further enrich this system by partitioning our variables $a_{1,\dots,m}$ into a subset of fixed inputs $a^{in}_{1,\dots,k}$, and variable outputs $a^{out}_{k+1,\dots,m}$. The system can then be framed functionally:

$$a^{out}_{k+1,\dots,m} = \mathcal{S}(a^{in}_{1,\dots,k}, M), \qquad \text{for } 1 \le k \le m. \tag{1}$$

This formulation can be used to capture Sudoku, an example used extensively in this work, where $a^{in}$ represents the input cells of a given Sudoku board, $a^{out}$ represents the cells that we aim to solve for, and $M$ encodes the rules of Sudoku.

MAXSAT Solvers can be leveraged to solve a broader class of problems that we refer to here as *Visual MAXSAT Problems*. These entail a MAXSAT problem where the inputs $a^{in}$ must first be derived from some other representation $a^{in}_{visual}$. This essentially results in a two-step training problem for which neurosymbolic architectures are optimised[1].

We have now established the preliminaries necessary to describe the Symbol Grounding Problem in the context of Visual MAXSAT solvers. It is the problem of identifying $a^{in}$ given only $a^{in}_{visual}$ and $a^{out}$. This motivates the distinction between two types of Visual MAXSAT Datasets: *grounded* and *ungrounded*. An ungrounded dataset contains $a^{in}_{visual}$ as data and $a^{out}$ as labels, while a grounded dataset additionally contains $a^{in}$ in its labels (See Figure 2.

Figure 2: Examples of Grounded and Ungrounded Visual MAXSAT Datasets, focusing on a $3 \times 3$ portion of a larger Sudoku board. Blue entries represent input cells. In previous work, SATNet is able to solve only the grounded version of the problem.

We note that it is significantly more difficult to solve the ungrounded version of a Visual MAXSAT problem, as training cannot be trivially broken up into two stages. It is this the class of problems that we tackle in this work.

### 2.2 Logical Constraint Solvers & SATNet

There has been significant recent interest in architectures that can integrate symbolic reasoning layers within neural networks. Many approaches, however, are only capable of integrating pre-existing logical constraints into these models [15–19]. In the context of our formalism, this is analogous to having a fixed set of clauses $M$ for a particular problem. Conversely, there exists a family of approaches that are not differentiable, but are able to learn logical constraints by example [9, 20, 21]. SATNet, however, sits somewhere in between these approaches, as it is both differentiable and able to learn a matrix $M$ in order to fit some input data [7]. There are a few other algorithms in this class, such as OptNet and $\partial$-Explainer [11, 12, 22].

---

[1]While it is also possible to train a system end-to-end to derive $a^{out}$ directly from $a^{in}_{visual}$, we argue that internally the system would need to have some form of representation of this two-step approach regardless.

### 2.3 Self-Supervised Pre-Training

Self-supervised pre-training has a long history in machine learning, notably being used to navigate highly non-convex loss landscapes in Deep Belief Networks (DBNs) [23, 24]. More recently, better methods for end-to-end training have emerged and self-supervision has now been used to pre-train image tasks on large, cheap unlabeled datasets to obtain slightly better performance on supervised tasks [25, 26].

In our work we return to the insight that motivated the original use of self-supervision for DBNs. The Symbol Grounding Problem essentially represents significant non-convexity in the problem space – both symbol meanings and the way in which symbols interact with one another must be learned in parallel, with local optima existing for many combinations of prospective groundings. Self-supervised pre-training enables us to start training from a favorable position on this loss landscape.

### 2.4 Clustering Algorithms & InfoGAN

Data clustering is a long-standing and rich field of computer science [27]. We leverage clustering in our method in order to conduct self-supervised pre-training. While many clustering algorithms exist, for our purposes we choose to use InfoGAN, as it is able to cluster across the semantic dimension which we are interested in for MNIST with very high accuracy [28].

InfoGAN is a Generative Adversarial Network architecture which boasts disentangled, interpretable latent encodings [29]. It maximizes the mutual information between a subset of the noise fed into its generator, and the observation which the discriminator makes. It is thus able to cluster data according to several interpretable variables. In the case of MNIST, these include handwritten digit thickness, slant, and most useful to us, the actual digit shape. This latter property is what we aim to leverage in this work. Specifically, InfoGAN can cluster MNIST digits according to their numerical value with 95% accuracy in a completely unsupervised fashion [28].

### 2.5 Knowledge Distillation

Knowledge Distillation is a technique for training machine learning models to reach comparable performance at inference time to a larger reference model, or an ensemble of models [30–33]. While more complex distillation techniques exist, our work leverages the concept in one of its most basic forms – simply training a smaller model from a dataset generated by a larger one in order to drastically improve inference time.

## 3 Method

Our main contribution is a pre-training pipeline used to bootstrap the learning process such that SATNet can bypass the Symbol Grounding Problem. Overall our method entails the following steps.

1. **Clustering**: We first perform unsupervised clustering of the input data, and distill the knowledge of the clusters into a digit classifier.

2. **Self-Grounded Training**: We then employ a custom *Symbol Grounding Loss* to identify how clusters map to the labels we have in our training data. Once the grounding is learned, we freeze it and train the rest of the system.

3. **Proofreading**: We conclude with an optional *proofreading* step which trains an additional layer in the SATNet architecture while the rest remain frozen. This was found to slightly improve performance in all SATNet architectures tested.

Before diving in to each of these steps, we will formalize the composite visual understanding/logical reasoning problem. Assume we look at a single instance of a MAXSAT problem with $N$ variables which can fall into one of $K$ classes, where each of the $N$ variables is represented by an image of size $C \times H \times W$. Our input data is then a tensor $x \in \mathbb{R}^{N \times C \times H \times W}$, and our desired one-hot encoded output $y \in \mathbb{R}^{N \times K}$. Our digit classifier $\mathcal{D}$ takes input $x$ and returns output $\mathcal{D}(x) = \hat{y}_{in} \in \mathbb{R}^{N \times K}$. We feed this result into our SATNet layer $\mathcal{S}$ such that $\mathcal{S}(\hat{y}_{in}) = \hat{y}_{out} \in \mathbb{R}^{N \times K}$. For Ungrounded Visual Sudoku using MNIST, we have $N = 81$ (one MAXSAT variable for each cell of the $9 \times 9$ Sudoku board), $K = 9$ (digits 1 through 9), and $C \times H \times W = 1 \times 28 \times 28$ (MNIST images).

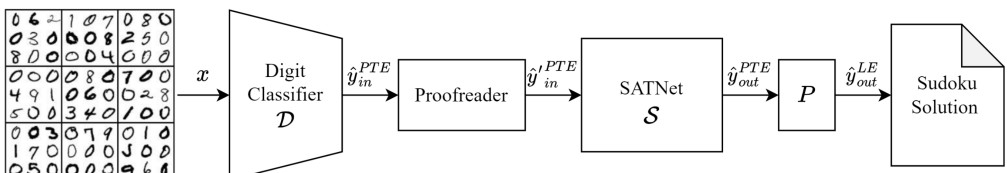

Figure 3: The architecture proposed in this work. It leverages self-supervised pre-training to solve Grounded Visual Sudoku, thereby overcoming the Symbol Grounding Problem affecting the original SATNet method.

## 3.1 Clustering

Our first step in solving an Ungrounded MAXSAT problem is identifying the patterns that exist in the input data. Intuitively, we do not have to start training a digit classifier from scratch when training composite visual reasoning architectures. There exists some semantic aspect of the input image which is of relevance to the MAXSAT problem at hand, and it can often be at least partially identified in a self-supervised setting. In the Visual Sudoku case, this entails clustering our images into 9 groups (corresponding values 1 through 9).

In our experiments, we use InfoGAN to perform the clustering, as it is capable of clustering MNIST digits with 95% accuracy [28]. Any clustering algorithm may be used here however, even ones that are not differentiable. Once the clustering is complete, we can distill the clustering knowledge back into a differentiable digit classifier. In our case, we generate a dataset using the clustering algorithm, and train LeNet on the cluster allocations of the training data [34]. By doing this we implicitly map each cluster onto some one-hot representation within $\hat{y}_{in}$. However, this one-hot encoding of the MAXSAT variables may not match with the encoding present in the labels $y$. We deal with this next.

## 3.2 Self-Grounded Training

While our clustering algorithm might be able to achieve high accuracy, it doesn't have any information about which numerical digit each cluster is actually associated with, since we don't have access to input cell labels. This is the crux of the Symbol Grounding Problem. In an Ungrounded MAXSAT setting, the only way to learn the association between digits and numerical clusters involves *jointly* learning the MAXSAT problem. In the case of Sudoku, this means that we must solve for the rules of puzzle and learn what each digit means simultaneously.

To reason about this, we consider two sets of encodings for digits: the pre-trained encoding (PTE) and the (correct) label encoding (LE), which we notate using superscripts. The digit classifier from the previous step outputs PTE-encoded predictions $\hat{y}_{in}^{PTE}$. There exists some unknown permutation matrix $P \in \mathbb{R}^{K \times K}$ that translates between encodings via $\hat{y}_{in}^{PTE} P = \hat{y}_{in}^{LE}$. Our goal is to align the PTE encoding with the LE encoding, so that we can make use of the training labels. The question of performing this translation before or after the SATNet layer is irrelevant, however. This is because the MAXSAT CNF clauses which SATNet implements are permutation-invariant [35]. This means that as long as supervision is provided correctly, we can train the SATNet layer $\mathcal{S}$ on either $\hat{y}_{in}^{PTE}$ or $\hat{y}_{in}^{LE}$. [2] In our approach we pass the prior through SATNet, and are left with $\hat{y}_{out}^{PTE}$ predictions.

We learn the correct permutation (without access to any of the input labels) simultaneously with training the SATNet layer, by introducing a *Symbol Grounding Loss*, which is intended to be a smooth function that is minimized when $\hat{y}_{out}^{PTE} P \approx y^{LE}$ for some permutation matrix $P$.

Note that $\hat{y}_{out}^{PTE}$ and $y^{LE}$ are $N \times K$ matrices, and let $\hat{y}_{out}^{PTE}(i)$ and $y^{LE}(i)$ denote the $i$th columns of these matrices. Then, $y^{LE}(i)$ is a 1-hot vector capturing the entries of the output that are labeled $i$ (in the correct label encoding), while $\hat{y}_{out}^{PTE}(i)$ is a vector of predicted probabilities that the output is

---

[2]While this is expected, this was not explicitly stated in the original SATNet paper. We were able to verify this empirically by applying any permutation on the one-hot encodings of the digits in the nonvisual Sudoku setting and SATNet's performance is identical even without re-training.

labeled $i$ (in the pre-trained label encoding). We define the following loss $\mathcal{L}$:

$$\mathcal{L}(\hat{y}_{out}^{PTE}, y^{LE}) := 1 - \mathrm{mean}_i(\max_j(\exp[-\mathrm{BCE}(y^{LE}(j), \hat{y}_{out}^{PTE}(i))])), \tag{2}$$

where $\mathrm{BCE}(\cdot, \cdot)$ denotes the binary cross-entropy loss between two vectors:

$$\mathrm{BCE}(v, w) = -\frac{1}{n}\left(\sum_{k=1}^{n} v_k \log(w_k) + \sum_{k=1}^{n}(1 - v_k)\log(1 - w_k)\right).$$

**Proposition 3.1.** *Suppose $\mathcal{L}$ is defined as in (2). Then:*

1. *$\mathcal{L}(\hat{y}_{out}^{PTE}, y^{LE})$ is minimized if and only if $\hat{y}_{out}^{PTE}P = y^{LE}$ for some permutation matrix $P$.*

2. *In this case, the matrix $P$ is given by $P_{ij} := \exp[-\mathrm{BCE}(y^{LE}(j), \hat{y}_{out}^{PTE}(i))]$.*

This proposition, which is proven in Appendix A, shows that by minimizing $\mathcal{L}$, we learn an approximate permutation matrix $\hat{P} \approx P$, given by:

$$\hat{P}_{ij} := \exp[-\mathrm{BCE}(y^{LE}(j), \hat{y}_{out}^{PTE}(i))]. \tag{3}$$

In practice, we do not minimize $\mathcal{L}$, since the $\max$ function presents an obstacle to effective training. Therefore, we relax the $\max$ operation to a function $\mathrm{approxmax}$. This finally gives us our *Symbol Grounding Loss* $\mathcal{L}_{SG}$:

$$\mathcal{L}_{SG}(\hat{y}_{out}^{PTE}, y^{LE}) := 1 - \mathrm{mean}_i(\mathrm{approxmax}_j(\exp[-\mathrm{BCE}(y^{LE}(j), \hat{y}_{out}^{PTE}(i))])). \tag{4}$$

In our experiments, we set $\mathrm{approxmax}$ equal to the 2-norm; however, we did not find that performance was sensitive to the exact choice of $\mathrm{approxmax}$, and other choices are also reasonable.

Having defined $\mathcal{L}_{SG}$, we incorporate it into our training pipeline as follows: First, we freeze the digit classifier $\mathcal{D}$, and train $\mathcal{S}$ under $\mathcal{L}_{SG}$. This begins to train $\mathcal{S}$ while also learning a permutation matrix $\hat{P} \approx P$ (defined by (3)). Note that since we are working with the Ungrounded Visual Sudoku task, the permutation matrix is learned by means of SATNet itself, and it is impossible for labels to be leaked, since the training process does not even have access to labels for the input entries.

Second, once $\hat{P}$ has converged to a clear permutation matrix, we freeze this permutation and use it to align the PTE labels with the correct LE labels by multiplying the final outputs $\hat{y}_{out}^{PTE}$ by the learned $\hat{P}$. Now that the Symbol Grounding Loss is no longer needed, we switch to the traditional BCE loss and complete the training of $\mathcal{S}$, also unfreezing $\mathcal{D}$ to allow additional training.

### 3.3 Proofreading

The performance of a SATNet architecture can be improved by the addition of a *Proofreader* layer. This consists of a linear layer added just before the SATNet layer $\mathcal{S}$, initialized to a slightly noisy identity transform $\mathbb{R}^{N \times K} \to \mathbb{R}^{N \times K}$. (In the Sudoku case, $N = 81$ and $K = 9$.) We freeze the layers in the original model, and train only the proofreader layer. This is an optional final step resulting in a slight performance improvement. We find that the Proofreader layer also improves the performance of the original SATNet (with label leakage), in both the visual and nonvisual Sudoku settings.

## 4 Results

The above procedure allows us to achieve comparable results on an Ungrounded Visual Sudoku Dataset as the original SATNet architecture has in the grounded setting[3], with results being presented in Table 1. We may thus claim to solve the Symbol Grounding Problem in the case of Visual Sudoku.

All experiments were carried out on a Nvidia GTX1070 across 100 epochs, with each epoch taking roughly 2 minutes. The Adam optimiser was used with learning rate of $2 \times 10^{-3}$ for the SATNet layer, and $10^{-5}$ for the digit classifier [36]. Standard deviations were calculated across 5 runs. We used the Sudoku Dataset made available under an MIT License from the original SATNet work [7].

---

[3]Note that training under a grounded dataset is equivalent to the label leakage problem described in [8]

| Model Configuration | Grounded vs. Ungrounded Data | Total Board Accuracy (%) | Per-Cell Accuracy (%) | Visual Accuracy (%) |
|---|---|---|---|---|
| Original SATNet | grounded | $66.5 \pm 1.0$ | $98.8 \pm 0.1$ | $99.0 \pm 0.0$ |
| Original SATNet | ungrounded | $0 \pm 0.0$ | $11.2 \pm 0.1$ | $11.6 \pm 0.0$ |
| **Our Method** | **ungrounded** | **$64.8 \pm 3.0$** | **$98.4 \pm 0.2$** | **$98.9 \pm 0.1$** |

Table 1: Performance of our method compared to the original SATNet architecture between grounded and ungrounded versions of the Visual Sudoku problem. Note that we distinguish the total board accuracy (how many 81-cell boards are completely correct) from per-cell accuracy (how many board cells are correct) and visual accuracy (how many input board cells are correct). Our method achieves comparable performance on a significantly more difficult version of the problem, thus solving the Symbol Grounding Problem.

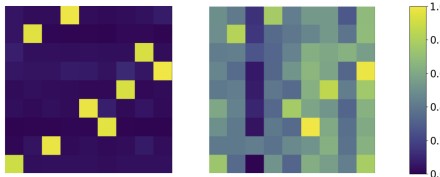

Figure 4: Permutation matrices extracted from the Symbol Grounding Loss function. On the left is a matrix extracted given a clustering with high accuracy, and the right matrix shows the results in a case where the clustering accuracy was below the necessary threshold (see Section 4.1).

During our pre-training pipeline, the clustering step achieves $95.6 \pm 0.4\%$ clustering accuracy. Under the Symbol Grounding Loss, our self-grounded training achieves $22.3 \pm 1.0\%$ per-cell accuracy. One thing to note is that the self-grounded training step is susceptible to overfitting, and one needs to employ early stopping on the basis of per-cell error in order to learn the permutation matrix $\hat{P}$. See Figure 4 for an example of a learned $\hat{P}$ matrix.

Note that it is expected that the ungrounded version of the dataset will produce slightly worse results since it carries less information in its labels than its grounded counterpart. Another relevant aspect is that InfoGAN itself is sensitive to random seed. 4/10 runs converge to a clustering below the threshold necessary to ground symbols. We discuss this limitation further in Section 4.1.

## 4.1 Effect of Clustering Accuracy

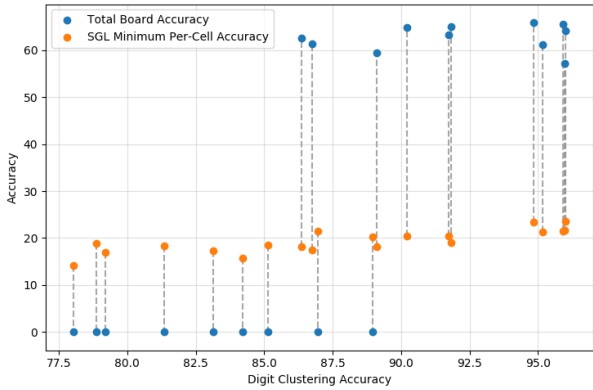

Figure 5: The effect of InfoGAN's clustering accuracy on our method's total board accuracy (blue) and per-cell accuracy during the Symbol Grounding Loss training phase (orange). Each pair of points connected by a dashed line indicates a different experiment. We note the sharp performance drop at roughly 88% clustering accuracy.

An important ablation test to define some limitations of our approach is a study on the effect of clustering accuracy on our pre-training performance. It is difficult to measure this, as performance could vary based on the distribution of predictions across clusters, not only raw clustering accuracy. In this study we run our pipeline against InfoGAN at different stages of its training. In this way the cluster assignments start out uniform (based on noisy initialization) and gradually anneal to a $89.6 \pm 7.7\%$ accurate clustering. We find that our system requires roughly at least 88% clustering accuracy in order for the rest of the pipeline to progress. This is shown in Figure 5. While this is a notable limitation to our approach, solving Un-

grounded Visual Sudoku was not at all possible with SATNet prior to this work. Furthermore, a threshold of 88% accuracy is not nearly as high as one might naively expect. Given that our input dataset contains on average 36.2 input cells per board, 88% digit classification accuracy gives less than a 0.1% chance of identifying an input board state perfectly with the initial clustering.

## 4.2 Effect of Distillation

In the case that the clustering algorithm used in our first pre-training phase is differentiable, the distillation step becomes optional. Despite this, it is desirable to distill our clustering model if there exists some smaller architecture that can achieve similar performance in the supervised setting. This is the case with InfoGAN, which in its standard form uses an architecture with 7,307,997 parameters. We distill this into a LeNet-derived architecture [34], with only 1,049,080 parameters and comparable performance, as shown in Table 2. Training speed changes from $602 \pm 5$ seconds/epoch to $255 \pm 3$ seconds/epoch between the two architectures.

| Digit Classifier | Digit Clustering Accuracy (%) |
|:---:|:---:|
| InfoGAN | $89.6 \pm 7.7$ |
| Distilled LeNet | $86.2 \pm 13.5$ |

Table 2: The effect of distilling InfoGAN into a smaller LeNet-based convolutional architecture. InfoGAN performance has a tendency to plateau at different levels based on seed. Here we show performance across all runs, whereas successful ones are used in the downstream pipeline. A "successful" InfoGAN run will plateau at roughly 95% accuracy.

## 4.3 Effect of Proofreading

Proofreading improves the performance of both visual and non-visual Sudoku, as seen in Table 3. We achieve the following results by training the proofreader with ungrounded Datasets even if the original model which it augments was trained with the grounded version.

| Model Configuration | Proofreader Present? | Total Board Accuracy (%) | Per-Cell Accuracy (%) | Visual Accuracy (%) |
|:---:|:---:|:---:|:---:|:---:|
| Original Non-visual | no | $96.6 \pm 0.3$ | $\mathbf{99.9 \pm 0.0}$ | N/A |
| **Original Non-visual** | **yes** | $\mathbf{97.1 \pm 0.3}$ | $\mathbf{99.9 \pm 0.0}$ | N/A |
| Original Visual | no | $66.5 \pm 1.0$ | $\mathbf{98.8 \pm 0.1}$ | $\mathbf{99.0 \pm 0.0}$ |
| **Original Visual** | **yes** | $\mathbf{67.6 \pm 1.2}$ | $98.6 \pm 0.1$ | $\mathbf{99.0 \pm 0.0}$ |
| Our Method | no | $62.8 \pm 3.2$ | $\mathbf{98.6 \pm 0.1}$ | $\mathbf{98.9 \pm 0.1}$ |
| **Our Method** | **yes** | $\mathbf{64.8 \pm 3.0}$ | $98.4 \pm 0.2$ | $\mathbf{98.9 \pm 0.1}$ |

Table 3: The effect of adding a proofreading layer to the original versions of SATNet for both visual and non-visual Sudoku datasets, as well as the pre-training method proposed in this paper. We show that a proofreader uniformly improves the Total Board Accuracy of SATNet.

We note that the numbers above from the original architectures reflect our reproduction of the results in the original paper. Please see Appendix B for further details.

# 5 Discussion

## 5.1 Sensitivity of SATNet to Random Seeds

It was described in [8] that SATNet exhibits a high sensitivity to the choice of random seed. For instance, 8 out of 10 random seeds would fail even with label leakage. While we initially reproduced this behavior, such sensitivity can in fact be circumvented with a minor correction to the PyTorch implementation, detailed further in Appendix B. We use the corrected, stable model for comparison in all our results.

## 5.2 Incorrect Upper Performance Bound

In the original SATNet paper, it is argued that the performance of the visual Sudoku model is bound by the probability of identifying all the input cells on a particular board correctly. Thus when using LeNet, which has a classification accuracy of 99.2%, the best performance we can expect on our dataset with 36.2 input cells on average is $0.992^{36.2} = 74.8\%$ [34, 7]. This is not exactly accurate.

It is not necessarily true that the SATNet layer cannot solve a board correctly if some number of input cells are wrong. Intuitively, if one finds two of the same numbers as inputs in a row of a Sudoku puzzle, one can infer that one of those inputs might have been classified incorrectly. This can then be used to make an educated guess about the correct final board state. We are able to show that the SATNet layer is actually able to reason about incorrect input cells to a certain extent. Interestingly, SATNet's ability to reason is affected by whether an incorrectly labeled digit results in an unsolvable board or not. It is also affected by the presence of a Proofreader layer. Details on these experiments can be found in Appendix C.

While the upper bound posed originally may not be strictly correct, it is still a good guideline. Deriving a strict upper performance bound is likely quite difficult as the mathematics of logical problems such as Sudoku are not fully understood.

## 6 Limitations & Future Work

While our method is able to address a new class of Visual MAXSAT problems with SATNet, it is limited by the need to prime the digit classifier with correct data clusters (see Section 4.1). This imposes a constraint on which datasets can be used as visual inputs to this pipeline. One facet of this limitation is the fact that the current Symbol Grounding Loss function only supports inferring a permutation between the pre-trained encoding and the label encoding. This means that if there are $K$ label classes, the clustering algorithm must cluster the input data accurately in $K$ clusters. One might imagine allowing the Symbol Grounding Loss to support a more general surjective mapping between encoding domains, allowing for a higher number of clusters (and consequently a higher accuracy).

A second limitation is the tendency of the Symbol Grounding Loss to overfit somewhat quickly. While we experimented with several loss function formulations, further experimentation may prove useful. Implementation of regularisers in the architecture may also be an interesting avenue of research.

We believe neither of these limitations is fundamental; future investigation may help to alleviate them.

## 7 Societal Impacts

The goal of the present work is to advance methods integrating deep learning and logical reasoning, which has long been a goal of artificial intelligence and admits a broad range of applications. We also alleviate reproducibility issues with prior SATNet systems, describing in Section 5.1 how to fix previously observed training instabilities [8].

Potential negative implications from our work are largely indirect and hard to assess. We envision the possibility of work on neurosymbolic methods leading to unrealistic expectations of the power of deep learning methods, which are not yet capable of sophisticated reasoning. This could lead to inappropriate trust placed in current deep learning methods, or to backlash if expectations fall short.

## 8 Conclusion

Our work lays out a foundation for distinguishing between grounded and ungrounded variants of Visual MAXSAT problems, and presents a self-supervised pre-training methodology which enables SATNet to solve both classes. The ability to solve the more difficult Ungrounded Visual MAXSAT problems contrasts markedly with the previous state of the art, which was unable to surpass 0% accuracy on these tasks. Further, we describe a proofreading methodology which can be used to incrementally improve both our architecture and prior models. This work extends the current state of the art for logical constraint-learning neurosymbolic methods, a promising area of research which boasts the potential to dramatically broaden the range of problems which machine learning can address.

## Acknowledgments

We thank the anonymous reviewers for insightful comments. This work was supported, in part, by Individual Discovery Grants from the Natural Sciences and Engineering Research Council of Canada, and the Canada CIFAR AI Chair Program.

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
