# A  Proof of Proposition 3.1

In this section, we prove Proposition 3.1.

**Proposition.** *Suppose $\mathcal{L}$ is defined as in* (2). *Then:*

1. *$\mathcal{L}(\hat{y}_{out}^{PTE}, y^{LE})$ is minimized if and only if $\hat{y}_{out}^{PTE} P = y^{LE}$ for some permutation matrix P.*

2. *In this case, the matrix P is given by $P_{ij} := \exp[-\operatorname{BCE}(y^{LE}(j), \hat{y}_{out}^{PTE}(i))]$.*

*Proof.* We first consider part (1), recalling that:

$$\mathcal{L}(\hat{y}_{out}^{PTE}, y^{LE}) := 1 - \operatorname{mean}_i(\max_j(\exp[-\operatorname{BCE}(y^{LE}(j), \hat{y}_{out}^{PTE}(i))])).$$

Since the BCE loss is minimized at 0, we have:

$$\mathcal{L}(\hat{y}_{out}^{PTE}, y^{LE}) = 1 - \operatorname{mean}_i(\max_j(\exp[-\operatorname{BCE}(y^{LE}(j), \hat{y}_{out}^{PTE}(i))]))$$
$$\geq 1 - \operatorname{mean}_i(\max_j(\exp[0]))$$
$$= 0.$$

and equality holds if and only if $\max_j(\exp[-\operatorname{BCE}(y^{LE}(j), \hat{y}_{out}^{PTE}(i))]) = 1$ for every $i$. This statement is true if and only if for every $i$ there exists a $j$ such that $\exp[-\operatorname{BCE}(y^{LE}(j), \hat{y}_{out}^{PTE}(i))] = 1$, or equivalently such that $\operatorname{BCE}(y^{LE}(j), \hat{y}_{out}^{PTE}(i)) = 0$.

Therefore, $\mathcal{L}$ reaches its minimum at 0 if and only if for every $i$ there exists a $j$ such that $y^{LE}(j) = \hat{y}_{out}^{PTE}(i)$, proving part (1).

We now prove part (2). Suppose that $\hat{y}_{out}^{PTE} P = y^{LE}$, and suppose that $\sigma$ is the permutation defined by $P$, so that $\sigma(i^{PTE}) = j^{LE}$. Then, for each $i, j$, we have:

$$\operatorname{BCE}(y^{LE}(j), \hat{y}_{out}^{PTE}(i)) = \begin{cases} 0 & \text{if } \sigma(i) = j \\ +\infty & \text{otherwise,} \end{cases}$$

and therefore

$$\exp[-\operatorname{BCE}(y^{LE}(j), \hat{y}_{out}^{PTE}(i))] = \begin{cases} 1 & \text{if } \sigma(i) = j \\ 0 & \text{otherwise.} \end{cases}$$

This proves part (2). $\qquad \square$

# B  Random Seed Sensitivity Fix

There are two aspects of the implementation which alleviate the sensitivity of SATNet on random seed. Please note that for this section we discuss the SATNet architecture from the original paper trained on an Grounded Visual Sudoku dataset [7]. The first was a minor bug in the CUDA implementation of SATNet's backprop calculation. The second relates to how supervision is provided on the digit classifier $\mathcal{D}$ during training.

Recall in section 2.1 that we can partition any Grounded Visual MAXSAT labels into input and output variable labels. Let us also reference our notation from section 3, where we have our digit classifier $\mathcal{D}(x) = \hat{y}_{in}^{PTE}$ and our SATNet layer $\mathcal{S}(\hat{y}_{in}^{PTE}) = \hat{y}_{out}^{PTE}$. We essentially have two options for returning the architecture's predictions for the input variables, as they are present in both $\hat{y}_{in}^{PTE}$ and $\hat{y}_{out}^{PTE}$. The choice of which of these to return results in a significant performance difference. The original SATNet model returned the input variable predictions from $\hat{y}_{out}^{PTE}$, whereas we return the ones in $\hat{y}_{in}^{PTE}$. Note that since the input variables are held constant in the SATNet layer $\mathcal{S}$, the nominal value of the input variables is the same between these two returns. The only difference is the implication of how gradients are computed.

# C Effect of Error Injection for Visual Sudoku

We construct an experiment by running the nonvisual Sudoku model, and perturbing input cell labels in the test datasets. Under no change to the original architecture, a SATNet layer trained on correct inputs is able to solve some small percentage of the boards with erroneous inputs. Interestingly, SATNet's ability to solve these puzzles depends on whether the error injection resulted in an unsolvable board or not. While adding a proofreader improves performance under normal circumstances, in the presence of injected errors it worsens performance.

| Number of Injected Input | Total Board Accuracy (%) | |
| :---: | :---: | :---: |
| Cell Errors per Board | Solvable | Unsolvable |
| 0 | $96.6 \pm 0.3$ | $96.6 \pm 0.3$ |
| 1 | $3.2 \pm 2.0$ | $0.0 \pm 0.0$ |
| 2 | $0.1 \pm 0.0$ | $0.0 \pm 0.0$ |
| 3 | $0.0 \pm 0.0$ | $0.0 \pm 0.0$ |

Table 4: Solvable/Unsolvable split under error injection for traditional nonvisual Sudoku solver. Some boards are still solved by SATNet even when not all input cells are correct.

| Number of Injected Input | Total Board Accuracy (%) | |
| :---: | :---: | :---: |
| Cell Errors per Board | Solvable | Unsolvable |
| 0 | $97.1 \pm 0.3$ | $97.1 \pm 0.3$ |
| 1 | $1.9 \pm 0.0$ | $0.0 \pm 0.0$ |
| 2 | $0.0 \pm 0.0$ | $0.0 \pm 0.0$ |
| 3 | $0.0 \pm 0.0$ | $0.0 \pm 0.0$ |

Table 5: Solvable/Unsolvable split under error injection for nonvisual Sudoku solver with proofreader.

# D Codebases used for Experimentation

Our experiments are built on top of two codebases. We leverage the SATNet implementation provided by the original authors at `https://github.com/locuslab/SATNet`, in addition to an InfoGAN implementation available at `https://github.com/Natsu6767/InfoGAN-PyTorch`. We have made our implementation public at `https://github.com/SeverTopan/SATNet`.