# OpenReview forum: "Techniques for Symbol Grounding with SATNet"
_NeurIPS.cc/2021/Conference — NeurIPS 2021 Spotlight_

### Official Review · Reviewer_bNAF · 2021-07-02

**Rating:** 6
**Confidence:** 3

**Summary:**

This work focuses on the problem of symbolic grounding in order to solve a sudoku task with perceptual input. The proposed method extends SATNet, a previous work based on the differentiable MAXSAT solver, and introduces several improvements that enable the model to solve the task without label leakage. Experiments are conducted on a visual sudoku dataset, demonstrating a superior performance of the proposed method.

**Limitations And Societal Impact:**

The limitations and societal impact of this paper is sufficiently discussed.

**Main Review:**

The incorporation of neural and symbolic methods in solving end-to-end tasks has been an important yet challenging direction of modern AI. This work contributes to this direction by tackling the symbolic grounding problem on an ungrounded visual sudoku task, on which previous methods fail completely. The experiments are sufficiently conducted to assess the effectiveness and limitations of the proposed method. The paper is very well written.

There are a few concerns and questions I would like to raise in this review:

1. In the problem setting, each input sudoku is automatically cut into 81 pieces into single-digit images, this throws away the "geometric" information of the game and drastically simplifies the perceptual module. I wonder if there is going to be any improvements if this geometric information is coupled to the learning of the game rules, as they are strongly related in reality.

2. Another direction of grounding symbols to perceptual inputs is via reinforcement learning, which has been demonstrated in the visual question answering task [1][2]. A discussion on the relationship between these approaches will contribute to the strength of this paper.

3. One can imagine the optimization during the self-grounded training is very sensitive to the structure of the underlying rules (or are they?). Working with one example of visual sudoku is insufficient to demonstrate the generality of this approach.

Minor: Line 99 end, close parenthesis.

In general, I think this is a well-written paper with solid results, but I also find the technical novelty of this work is slightly below the bar of this conference (since this work builds on top of an existing model and it is unclear how well the pretraining scheme can be adopted in other models and tasks). My score on this paper is "weak accept".

[1] Mao et al. "The Neuro-Symbolic Concept Learner: Interpreting Scenes, Words, and Sentences From Natural Supervision", ICLR 2019
[2] Yi et al. "Neural-Symbolic VQA: Disentangling Reasoning from Vision and Language Understanding", NeurIPS 2018

Post rebuttal:
I thank the authors for their efforts in addressing the comments from the reviewers. Even though the technical complexity of the task is limited and more work is needed to further establish the merit of the full method, I feel this is still a good paper to be presented in NeurIPS. My final assessment of this paper remains "weak accept".

**Time Spent Reviewing:**

2.5

---

> ### Author Response · Authors · 2021-08-10
> **Responses to the Questions Posed**
>
> We would like to thank the reviewer for their review and informative suggestions. Please find our responses to the main points below.
>
> > 1. In the problem setting, each input sudoku is automatically cut into 81 pieces into single-digit images, this throws away the "geometric" information of the game and drastically simplifies the perceptual module. I wonder if there is going to be any improvements if this geometric information is coupled to the learning of the game rules, as they are strongly related in reality.
>
> It is correct that we encode the geometric information into our model. Each Sudoku cell is treated separately, and more generally each Visual MAXSAT variable would need to be input separately with our current framework. What is being proposed is a very interesting extension. Tackling this problem would require the model to not only learn how to reason about visual sudoku digits, but also how to delineate where each digit actually starts and ends, and which digits are input vs. output variables. This would be analogous to providing zero knowledge of the Visual MAXSAT problem being proposed. We think this would be a difficult and interesting challenge.
>
> > 2. Another direction of grounding symbols to perceptual inputs is via reinforcement learning, which has been demonstrated in the visual question answering task [1][2]. A discussion on the relationship between these approaches will contribute to the strength of this paper.
>
> Thank you for the references, these would indeed be interesting to discuss and we will add them into our analysis of prior work. The topic of Visual Question Answering is particularly interesting since [3] has been made available. From a quick review, one of the interesting aspects of SATNet is that as long as the problem at hand is solvable by MAXSAT, no DSL has to be pre-coded in order for the model to learn. That said, we would have to think further about how to apply this work to something like the CLEVR Dataset, and how the differences in the Symbol Grounding problems presented by the RL-inspired works relate to ours.
>
> [3] Sihyun Yu, Sangwoo Mo, Sungsoo Ahn, and Jinwoo Shin. Abstract Reasoning via Logic-guided Generation. _arXiv e-prints_, page arXiv:2107.10493, July 2021.
>
> > 3. One can imagine the optimization during the self-grounded training is very sensitive to the structure of the underlying rules (or are they?). Working with one example of visual sudoku is insufficient to demonstrate the generality of this approach.
>
> We have used the running example of Visual Sudoku mainly due to time and resource constraints. In theory the self-grounded training generalizes to any problem fitting the parameters of a “Visual MAXSAT Problem,” which we introduce in section 2.1. We hope to have convinced our readers that our framework is theoretically sound, and backed it with empirical results and ablation tests in the Visual Sudoku context.

---

### Official Review · Reviewer_Aqg2 · 2021-07-14

**Rating:** 7
**Confidence:** 4

**Summary:**

This paper proposes to solve composite visual reasoning problems represented by ungrounded visual Sudoku through augmenting SATNet with several extra learning stages. A differentiable digit classifier is first obtained through distilling unsupervised clustering, which produces quite accurately clustered but potentially classes-permuted classification results. A Symbol Grounding Loss is developed to align the correspondingly permuted prediction from SATNet with ground truth classes, while also supervising the reasoning result. The experiments on ungrounded visual Soduku demonstrate that the additional training steps substantially improved the performance. It is also shown that a proofreader, neural network that finetunes the digit classification before SATNet, further improves the performance.

**Limitations And Societal Impact:**

The authors have discussed several limitations throughout the paper. However, one additional critical limitation remains unaddressed: Since the clustering step is task agnostic, it might not produce the symbols required by the task. For example, what if the rule is based on the handwriting style of the digit rather than the class of the digit? Or both of them matter? It seems hard for the later end-to-end training with the distilled differentiable classifier to adjust such a mismatch.






**Main Review:**

### Originality

To the best of my knowledge, the proposed distilling clustering result as pre-trained classifier, Symbol Grounding Loss, and proofreading are novel.

### Quality

The methods presented make sense intuitively and are supported by prior works or theoretical results. Experiments demonstrate substantial improvement upon prior SATNet results over ungrounded Soduku. However, one of my main concerns is the potential label leaking via $y^{LE}$:
-  To avoid label leaking, $y^{LE}$ should not contain labels for input variables, i.e. the digits given on the board, during both permutations matrices training and normal supervised training. However, it is not clear from the text that whether this is the case and how are these input variables are handled in the Symbol Grounding Loss and normal BCE loss. Are they just omitted from the term?
- In Appendix B, it is noted that the gradient computation would be different by outputting the input variable predictions using $ y_{out}^{PTE} $ versus $ y_{in}^{PTE} $ (there should be hat on y). But if the input variable predictions are not supervised at all in the ungrounded visual Suduku, why would the gradient computation change?

Without clarifying this, it is unclear that the main claim about solving ungrounded composite visual reasoning problems is supported.

### Clarity

The paper is clearly written overall, but the experiment section has some details missing.

- On line 233 and figure 5, it is reported that the per-cell accuracy with self-grounding training is about 20%. This number seems very low compared to the final 98.4% per-cell accuracy reported in table 1 and the high clustering accuracies. Why is this the case? Also, why does the per-cell accuracy barely increases as the clustering accuracy increases?
- The InfoGAN clustering accuracy is reported as $95.6 \pm 0.4 $% on line 232, but $89.6 \pm 7.7 $% is reported in table 2 and line 255. Are the settings different?
 - In figure 5, is the clustering accuracy referring to InfoGAN accuracy or distilled LeNet accuracy? I assume that LeNet distillation is used as the real digit classifier when performing the corresponding self-grounded training.
- Distilled LeNet is reported to have a quite large variance in table 2. Does the distillation step incur variance? i.e. for single trained InfoGAN, what is the variance of the LeNet training trials.
- Since InfoGAN and Distilled LeNet performance affect the symbol grounding quality drastically, are the results in tables 1 and 3 produced by using a selected run of InfoGAN and LeNet? Or are their variance also accounted for?
- In table 6 (the updated version of table 3 in the appendix), the bolding seems incorrect for per-cell accuracy for the last two rows.
-  The criteria for early stopping the self-grounded training, which seems very important for the pipeline to perform well, should be explained more concretely.

### Significance
This work presents a novel and effective approach to jointly perform reasoning and symbol grounding, which is an important mark of intelligence. The exact training procedure seems relatively fragile with the need to select lucky runs of clustering and early stop self-grounded learning training. Nonetheless, this work demonstrates a working pipeline that future works can improve upon.


**Time Spent Reviewing:**

15

---

> ### Author Response · Authors · 2021-08-10
> **Clarifications on the Issues Raised**
>
> We would like to thank the reviewer for their thorough analysis of our work. We hope to address the issues raised below.
>
> > 1. To avoid label leaking, $y^{LE}$ should not contain labels for input variables, i.e. the digits given on the board, during both permutations matrices training and normal supervised training. However, it is not clear from the text that whether this is the case and how are these input variables are handled in the Symbol Grounding Loss and normal BCE loss. Are they just omitted from the term?
>
> Our claim is that by definition, the way that we have set up the Ungrounded Visual Sudoku problem (without $a_{in}$, see Figure 2) makes label leakage impossible, since the input cell labels aren’t present at all in the dataset made available to the training algorithm. The y-hat’s discussed in section 3 are made available for both input and output cells, but supervision is only provided on the output cells in the ungrounded context. We discuss this in section 2.1.
>
> > 2. In Appendix B, it is noted that the gradient computation would be different by outputting the input variable predictions using $y_{out}^{PTE}$ versus $y_{in}^{PTE}$ (there should be hat on y). But if the input variable predictions are not supervised at all in the ungrounded visual Suduku, why would the gradient computation change?
>
> In ungrounded visual sudoku, gradients actually still flow from output variables to the input variables through the SATNet layer. This is the means by which our visual accuracy grows from 95.6 to 98.9 during the final training step. This is actually what we think is one of the interesting aspects of the Symbol Grounding Problem. Essentially the identities of the handwritten digits are learned indirectly through the context provided by the rules of the puzzle and the output labels (we discuss this around line 178). It is very promising that SATNet is able to handle this level of indirection, as it is incredibly difficult, even for humans [1].
>
> [1] Oscar Chang, Lampros Flokas, Hod Lipson, and Michael Spranger. Assessing SATNet's ability to solve the symbol grounding problem. In H. Larochelle, M. Ranzato, R. Hadsell, M. F. Balcan, and H. Lin, editors, _Advances in Neural Information Processing Systems (NeurIPS)_, 2020.
>
> > 3. On line 233 and figure 5, it is reported that the per-cell accuracy with self-grounding training is about 20%. This number seems very low compared to the final 98.4% per-cell accuracy reported in table 1 and the high clustering accuracies. Why is this the case? Also, why does the per-cell accuracy barely increases as the clustering accuracy increases?
>
> We would like to first address the discrepancy between the self-grounded training and the final per-cell accuracy of our proposed framework. The primary goal of the self-grounded training is to train the model until a permutation matrix P can be learned (see Figure 4). This is where Symbol Grounding occurs. Once P is learned, the system can continue to learn under a traditional loss function.
>
> Originally we were hoping to complete training entirely under the Symbol Grounding Loss (SGL), but found that the formulation is not as effective at communicating gradients for the actual LeNet and SATNet models as Binary Cross-Entropy (BCE). This is why accuracy plateaus under SGL before BCE is introduced.
>
> Our intuition for why the per-cell accuracy only slightly increases with clustering accuracy is that qualitatively, “unsuccessful” clustering runs of InfoGAN are still mostly correct. It's usually one or two clusters which are mismatched (4 and 9 are usual suspects). Thus the symbol grounding loss still has similar nominal loss values, but the permutation matrix P ends up noisy in these cases, so further training under BCE is unsuccessful.
>
> > 4. The InfoGAN clustering accuracy is reported as 95.6±0.4 % on line 232, but 89.6±7.7 % is reported in table 2 and line 255. Are the settings different?
>
> On line 232, we express the InfoGAN clustering accuracy across successful runs only, while table 2 shows clustering accuracy across all runs. Thank you for pointing this out, and we should certainly clarify this.
>
> The idea of using selected runs has been brought up in several places in this review, so we would like to defend this decision here. What we have done in this paper is introduce a framework for solving the Symbol Grounding problem for Visual MAXSAT problems using SATNet. The clustering algorithm used is a smaller component of this framework, and the variance observed in InfoGAN is a limitation of InfoGAN as opposed to the framework itself. We discuss the framework’s sensitivity to clustering accuracy in our ablation study in section 4.1. InfoGAN can be transparently replaced with a different clustering algorithm to solve this variance issue (such as VQGAN’s, which another reviewer suggested).
>
> Furthermore, InfoGAN’s loss can be used as an indicator of whether it is settling on a “correct” clustering of digits or not across runs, hence there exists a signal that can be monitored during training in order to identify whether we are in fact learning a correct clustering.
>
> > 5. In figure 5, is the clustering accuracy referring to InfoGAN accuracy or distilled LeNet accuracy? I assume that LeNet distillation is used as the real digit classifier when performing the corresponding self-grounded training.
>
> This is a good point, we should definitely clarify this. The x-axis in this figure shows InfoGAN clustering accuracy. And it is correct that the self-grounded training uses the distilled LeNet model.
>
> > 6. Distilled LeNet is reported to have a quite large variance in table 2. Does the distillation step incur variance? i.e. for single trained InfoGAN, what is the variance of the LeNet training trials.
>
> The results shown for the Distilled LeNet are shown across all runs of InfoGAN, including ones which are “unsuccessful”  (i.e. fail to learn the correct digit clustering). What generally happens is that InfoGAN runs which do not end up being “successful”  result in poorer generalization when distilled into LeNet.
>
> > 7. Since InfoGAN and Distilled LeNet performance affect the symbol grounding quality drastically, are the results in tables 1 and 3 produced by using a selected run of InfoGAN and LeNet? Or are their variance also accounted for?
>
> That is correct, selected runs are used in these tables (discussed further in point 4 above).
>
> > 8. In table 6 (the updated version of table 3 in the appendix), the bolding seems incorrect for per-cell accuracy for the last two rows.
>
> We bolded the rows on the basis of the total board accuracy, as we would consider it the stronger signal for success (indeed, it is the only signal reported in the original SATNet paper). However the reviewer is correct that the bolding becomes incorrect for per-cell accuracy. We will revise our formatting.
>
> > 9. The criteria for early stopping the self-grounded training, which seems very important for the pipeline to perform well, should be explained more concretely.
>
> We used early stopping on the basis of Per-cell accuracy, which we considered to be a fairly standard approach in the Machine Learning community. For this reason we chose to not dive into detail, but we would be happy to elaborate this part if necessary in the next revision.
>
> > 10. This work presents a novel and effective approach to jointly perform reasoning and symbol grounding, which is an important mark of intelligence. The exact training procedure seems relatively fragile with the need to select lucky runs of clustering and early stop self-grounded learning training.
>
> We would like to briefly defend the robustness of our proposed framework. We would first argue that early stopping is not a serious point of fragility in our system, since there is a clear signal (per-cell accuracy) with which to perform early stopping. Furthermore, early stopping is a fairly standard technique for avoiding overfitting in ML. Secondly, while it is true that InfoGAN performance has high variance, we argue that this is more a limitation of InfoGAN as opposed to the framework we propose. This can be remedied with future improvements to the choice of clustering algorithm.
>
> In summary, we hope to have addressed the main concerns of the reviewer with this reply, particularly the issues raised about whether the main claims of the paper are supported. We argue that we are in fact solving the Symbol Grounding problem through our definition of Ungrounded Visual Sudoku, which is a significant contribution to SATNet architectures, and the field of Neurosymbolic AI in general. We hope that the reviewer would kindly reconsider our scoring.

---

> > ### Author Response · Authors · 2021-08-13
> > **Follow-up Comment on Accuracy Variance in Original SATNet Implementation**
> >
> > We would also like to briefly highlight that before our contributions, the results from the original SATNet implementation were subject to very significant variance. The original paper cited a 63.2% accuracy for the Grounded Visual Sudoku Task [1], but a follow-up paper was only able to reproduce 11.9±7.9% accuracy due to 8/10 runs leading to complete training failure [2].
> >
> > We not only contributed a solution to this instability (discussed in Appendix B), but also a solution for the ungrounded rendition of the same problem.
> >
> >
> >
> >
> > [1] Po-Wei Wang, Priya L Donti, Bryan Wilder, and Zico Kolter. SATnet: Bridging deep learning and
> > logical reasoning using a differentiable satisfiability solver. _International Conference on Machine Learning (ICML)_,
> > 2019.
> >
> > [2] Oscar Chang, Lampros Flokas, Hod Lipson, and Michael Spranger. Assessing SATNet's ability to solve the symbol grounding problem. In H. Larochelle, M. Ranzato, R. Hadsell, M. F. Balcan, and H. Lin, editors, _Advances in Neural Information Processing Systems (NeurIPS)_, 2020.

---

> > > ### Comment · Reviewer_Aqg2 · 2021-08-15
> > > **Response to the Reviewers**
> > >
> > > Thanks for the detailed response! It addresses most of my concerns, except question 2. Since it is central to the claim that the label leakage issue is not present, I'd like to make sure that my understanding of the task setup is full and accurate.
> > >
> > > It is still confusing to me why there would be any gradient flowing through the input variables predicted by SATNet, if it is not supervised at all (as replied to my question 1)? The SATNet should obtain the same loss by outputting anything for the input variable entries, like all zeros in figure 2.

---

> > > > ### Author Response · Authors · 2021-08-17
> > > > **Further Discussion on Input Cell Gradients**
> > > >
> > > > Absolutely, we'd be happy to clarify this as it's fairly subtle. As you suggest, since there are no labels available for the input cell predictions in the task, it appears strange that the model would be able to pass a gradient through to the digit classifier.
> > > >
> > > > We think the easiest way to think about how this might happen is to consider for a moment a perfectly trained instance of SATNet. Inside, it contains CNF clauses which perfectly encode the rules of Sudoku. We could then envision the following training scenarios:
> > > >
> > > >
> > > > $$
> > > > \underset{\text{Input Image}}
> > > > {
> > > > \begin{bmatrix}
> > > > \mathcal{0} & \mathcal{8} & \mathcal{0} \\\\
> > > > \mathcal{0} &  \mathcal{6} & \mathcal{0} \\\\
> > > > \mathcal{3} &  \mathcal{4} & \mathcal{0}
> > > > \end{bmatrix}
> > > > }
> > > > \overset{\text{Digit Classifier}}{\longrightarrow}
> > > > \underset{\text{Input Cell Predictions}}
> > > > {
> > > > \begin{bmatrix}
> > > > 0 & \color{blue} 8 & 0 \\\\
> > > > 0 & \color{blue} 6 & 0 \\\\
> > > > \color{blue} 3 & \color{blue} 4 & 0
> > > > \end{bmatrix}
> > > > }
> > > > \overset{\text{SATNet (Differentiable)}}{\longrightarrow}
> > > > \underset{\text{Output Cell Predictions}}
> > > > {
> > > > \begin{bmatrix}
> > > > 1 & \color{blue} 0 & 5 \\\\
> > > > 2 & \color{blue} 0 & 9 \\\\
> > > > \color{blue} 0 & \color{blue} 0 & 7
> > > > \end{bmatrix}
> > > > }
> > > > \underset{\text{Gradients}}{\overset{\text{Loss Function}}{\longleftrightarrow}}
> > > > \underset{\text{Output Cell Labels}}
> > > > {
> > > > \begin{bmatrix}
> > > > 1 & \color{blue} 0 & 5 \\\\
> > > > 2 & \color{blue} 0 & 9 \\\\
> > > > \color{blue} 0 & \color{blue} 0 & 7
> > > > \end{bmatrix}
> > > > }
> > > > $$
> > > >
> > > > Example 1. Correct input cell classification results in expected behavior. Consider the left-most matrix to represent the visual handwritten rendition of the inputs.
> > > >
> > > > $$
> > > > \underset{\text{Input Image}}
> > > > {
> > > > \begin{bmatrix}
> > > > \mathcal{0} & \mathcal{8} & \mathcal{0} \\\\
> > > > \mathcal{0} &  \mathcal{6} & \mathcal{0} \\\\
> > > > \mathcal{3} &  \mathcal{4} & \mathcal{0}
> > > > \end{bmatrix}
> > > > }
> > > > \overset{\text{Digit Classifier}}{\longrightarrow}
> > > > \underset{\text{Input Cell Predictions}}
> > > > {
> > > > \begin{bmatrix}
> > > > 0 & \color{blue} 8 & 0 \\\\
> > > > 0 & \color{blue} 6 & 0 \\\\
> > > > \color{blue} 3 & \color{red} 9 & 0
> > > > \end{bmatrix}
> > > > }
> > > > \overset{\text{SATNet (Differentiable)}}{\longrightarrow}
> > > > \underset{\text{Output Cell Predictions}}
> > > > {
> > > > \begin{bmatrix}
> > > > 1 & \color{blue} 0 & 5 \\\\
> > > > 2 & \color{blue} 0 & \color{red} 7 \\\\
> > > > \color{blue} 0 & \color{blue} 0 & \color{red} 4
> > > > \end{bmatrix}
> > > > }
> > > > \underset{\text{Gradients}}{\overset{\text{Loss Function}}{\longleftrightarrow}}
> > > > \underset{\text{Output Cell Labels}}
> > > > {
> > > > \begin{bmatrix}
> > > > 1 & \color{blue} 0 & 5 \\\\
> > > > 2 & \color{blue} 0 & 9 \\\\
> > > > \color{blue} 0 & \color{blue} 0 & 7
> > > > \end{bmatrix}
> > > > }
> > > > $$
> > > >
> > > > Example 2. Incorrect input cell classification results in downstream effects. Gradients flow through SATNet to indicate the misclassified digit.
> > > >
> > > > In Example 1, we show expected behavior of the model in the case where all input cells are classified correctly. The input predictions are then passed to SATNet, the CNF is applied, and the output cells are computed as expected. In the case where an input cell is misclassified such as in Example 2, however, the model finds itself in the interesting situation which we are discussing. The model must somehow propagate gradients through the SATNet layer to the digit classifier. It is able to do this with only knowledge of output cell labels because an input cell misclassification results in predictable downstream effects. In the example, the 9 could no longer appear as an output cell since each digit can only appear once per 3x3 block. There might also be non-local effects when taking account the rest of the board, as is shown by the 7 also changing its location in this made-up example. These effects create the gradients which influence the digit classifier: input cell identities (groundings) are learned contextually from their effects on the output cells.
> > > >
> > > > We can then relax the condition that SATNet has already been perfectly trained. The implication now is that at some point the model realizes that a given misprediction in the output cells is not caused by a misinterpretation of the rules of Sudoku, but an input cell misprediction. It would then pass the requisite gradients accordingly. To our knowledge, our work is the first demonstration of this.
> > > >
> > > > This effect is what motivates the pre-training framework which we introduce. During training there exists a delicate balancing act between the model realizing that it is mispredicting the rules of Sudoku versus the identities of the input digits. The clustering step in our framework bootstraps the digit classifier, and Self-Grounded training bootstraps SATNet. From there we show that standard training is sufficient to allow the system to train to completion.
> > > >
> > > > Please let us know if this answers your question. We would be happy to discuss further.

---

> > > > > ### Comment · Reviewer_Aqg2 · 2021-08-19
> > > > > **Further Discussion**
> > > > >
> > > > > Thanks again for the detailed answer and illustrations. I think we have some miscommunication about this question. Hopefully, the examples provided in your response can help clarify it. Unfortunately, I don't know how to draw the examples the same way; I'll try my best to refer to them.
> > > > >
> > > > > My question is why there is any gradient through the input variables **predicted by SATNet**, i.e. the blue zeros in the "output cell predictions matrix" in Example 1. As far as I understand, Appendix B claims that "We essentially have two options for returning the architecture’s predictions for the input variables, as they are present in both [the output of digit classifier & input of SATNet, i.e. the blue digits in "Input Cell Prediction" matrix] and [the output of the SATNet, i.e. the blue digits in "Output Cell Prediction" matrix]. The choice of which of these to return results in a significant performance difference...". It further notes that this is because the gradients are different, even though the nominal values of the two options are the same.
> > > > >
> > > > > This makes perfect sense to me for the grounded case: In this case, the blue zeros in Examples 1 should be replaced by the corresponding blue digits in the "Input Cell Prediction" matrix. Returning the blue digits in the "Input Cell Prediction" matrix from SATNet avoids the need to propagate the gradient through the SATNet layer, comparing to return the blue digits in the "Output Cell Prediction" matrix.
> > > > >
> > > > > But I wonder why this trick should make any difference for the ungrounded case? As shown in Example 1, the nominal values of the blue digits in the two matrixes are different: non-zeros in the "Input Cell Prediction" matrix and zeros in the "Output Cell Prediction" matrix. So here I assumed that the "Output Cell Prediction" matrix is a masked version of the full prediction of the SATNet, i.e. SATNet actually still predicts the values for the blue digits entries as described in Appendix B – "input variables are held constant in the SATNet layer". With this understanding, there should be no gradient through these predicted blue entries anyway, since they should not be supervised in the ungrounded case. So it shouldn't matter anymore which version of the blue digits is returned by SATNet; the SATNet could just return zeros for blue entries as shown in example 1.
> > > > >
> > > > > I am particularly concerned about this detail as it hints that there is some unseen supervision signal through these input digits still. Please let me know if I misunderstood anything, thanks!

---

> > > > > > ### Author Response · Authors · 2021-08-20
> > > > > > **Clarification on Typo**
> > > > > >
> > > > > > I think we may have found the source of confusion. There is a typo in our Appendix B. It should read "Grounded", not "Ungrounded" on lines 480 and 483. The sensitivity fix changes the way input cell labels are passed and is meant to solve accuracy variance in previous SATNet implementations, which were exclusively executed on grounded datasets. As you suggest, the trick doesn't make sense in the ungrounded case since there are no input cell labels present in the first place. Your intuition about the output cell masking is correct. Apologies for this, it is an editing mistake.

---

> > > > > > > ### Comment · Reviewer_Aqg2 · 2021-08-24
> > > > > > > **Reply to the Authors**
> > > > > > >
> > > > > > > That makes sense and sorry for not realizing that it could be an editing mistake in such a way. I will revise my score accordingly.

---

### Official Review · Reviewer_rhk1 · 2021-07-16

**Rating:** 5
**Confidence:** 2

**Summary:**

Advancement of deep leaning comes with several limitations such as lack of interpretability, adversarial attacks and difficulties in incorporating logical constraints. To alleviate these limitations, neurosymbolic AI has been studied that integrates neural networks with logical reasoning. One recent breakthrough is SATNet, which proposes a differentiable MaxSAT solver and finds its application in visual reasoning problems, such as learning rules of Sudoku solely from image examples.


This paper tackles the symbol grounding problem in neurosymbolic systems which is the inability to map visual inputs to symbolic variables without supervision (also known as label leakage). In this paper, authors argue that without any label leakage (that is, without any supervision in digit classification task), SATNet’s abilility to solve Sudoku drops to 0%. As a result, this paper proposes a self-supervised pre-training pipeline that enables SATNet to solve ungrounded MaxSAT problems, where label data are available only for output variables of the MaxSAT problem. The contributions of the papers are (i) proposing a self-supervised clustering and a distillation process to train a visual classifier within a SATNet architecture, (ii) introducing a loss function specific to symbol grounding problems that allows to train logical constraints on ungrounded symbol representation. Empirical evaluation shows a similar performance on ungrounded MaxSAT problem compared to grounded MaxSAT problem (which allows label leakage).


**Limitations And Societal Impact:**

Authors have discussed limitations and societal impact.

**Main Review:**

The paper is generally well written. The running example of Sudoku helps understand the technical contribution of the paper. Symbol grounding problem is well explained in the paper, specially in the context of Sudoku game.

The paper is somewhat incremental to previously proposed SATNet model. I am motivated that label leakage is a challenging problem in case of visual Sudoku where individual tasks such as image classification and logical reasoning can be separately solved successfully. However, without any label leakage, the problem is indeed difficult to solve. Thus, the paper would get attention from research community in neurosymbolic AI.


I am listing my concerns/questions to the authors in the following. I would be delighted to increase score if authors clarify my understanding.

1. The paper presents the main technical details with a running example of Sudoku. What other problems apart from Sudoku can be covered by the proposed integration to SATNet?

2. The paper introduces/discusses MaxSAT in a more abstract sense. How does SATNet solve Sudoku using MaxSAT? An illustration of MaxSAT formulation is missing in the paper, which arises the following question.
In line 155, authors mention that each of N variables represent an image of Sudoku. Are these variables Boolean? If so, how can a Boolean variable represent an image?

3. In line 172, LeNet is introduced without any reference.

4. What is the intuition behind proofreading layer? How does a slightly noisy identify transformation function benefit in performance?

5. In line 204, what is the precise definition of approxmax used in the experiment?

6. Why are experiments limited to Sudoku problem only? This limits the generalisation of the proposed formulation, at least empirically.

6. In table 1, for “grounded” version of SATNet, if per-cell accuracy is 99%, why is total-board accuracy only 67%? Moreover, why is total-board accuracy around 65% for both grounded and ungrounded cases? Any intuition behind that? Is this an average result? If so, how many different Sudoku samples are considered in the experiment?

7. In Figure 4, what is the interpretation behind different colors in both figures?

**Time Spent Reviewing:**

10

---

> ### Author Response · Authors · 2021-08-10
> **Responses to the Questions Posed**
>
> Thank you for the careful review. We hope to clarify some of the points brought up in our responses below. This review points to several edits which we will certainly include in a revision.
>
> > 1. The paper presents the main technical details with a running example of Sudoku. What other problems apart from Sudoku can be covered by the proposed integration to SATNet?
>
> The general class of problems which we are able to address is introduced in section 2.1 which we dub, “Visual MAXSAT Problems.” These problems can be solved by applying MAXSAT on a collection of symbols which are represented visually. With our current permutation matrix formulation, there also exists an implicit constraint that the output labels need to be in the same class domain as the inputs (for Sudoku, all input and output cells are one of 9 digits). We mention this in section 6, but we realize we should discuss this point more explicitly in section 2.1 of our next revision. It is also worth mentioning that Markov Logic Network (MLN) inference is essentially a MAXSAT problem, so our work can potentially enable MLN inference on visual inputs.
>
> Several other puzzle games similar to Sudoku fall into this category, for example Kakuro or Minesweeper. Other works [1, 2] have shown SATNet functioning in the context of something called the “MNIST Mapping” Problem and Raven’s Matrix problems.
>
> [1] Sihyun Yu, Sangwoo Mo, Sungsoo Ahn, and Jinwoo Shin. Abstract Reasoning via Logic-guided Generation. _arXiv e-prints_, page arXiv:2107.10493, July 2021.
>
> [2] Oscar Chang, Lampros Flokas, Hod Lipson, and Michael Spranger. Assessing SATNet's ability to solve the symbol grounding problem. In H. Larochelle, M. Ranzato, R. Hadsell, M. F. Balcan, and H. Lin, editors, _Advances in Neural Information Processing Systems (NeurIPS)_, 2020.
>
> > 2. The paper introduces/discusses MaxSAT in a more abstract sense. How does SATNet solve Sudoku using MaxSAT? An illustration of MaxSAT formulation is missing in the paper, which arises the following question. In line 155, authors mention that each of N variables represent an image of Sudoku. Are these variables Boolean? If so, how can a Boolean variable represent an image?
>
> We introduce the MAXSAT problem in section 2.1, and we discuss in section 2.2 how SATNet is able to solve the Visual Sudoku problem (albeit the grounded version).
>
> The N variables mentioned on line 155 refer to categorical variables which fall into one of K classes. These are then one-hot encoded into a collection of boolean variables which are made available to the MAXSAT solver. In the context of Visual Sudoku, we first have 81 images representing each cell in a given board (digits are assigned images using the MNIST database); these are then classified into one of 9 digits (1 through 9), and then one-hot encoded before being fed into SATNet. Intuitively speaking, 9 Boolean variables represent an image, if we imagine the one-hot encoding is then a relaxation of 9 boolean variables.
>
> > 3. In line 172, LeNet is introduced without any reference.
>
> Line 172 lists citation [34] for “Gradient Based Learning Applied to Document Recognition” by LeCun et al., which we believe is the original LeNet paper, unless we have mis-cited? The confusion might be due to the fact we choose to add citations at the end of a sentence, and we can move citations right after the name of each work.
>
> > 4. What is the intuition behind proofreading layer? How does a slightly noisy identify transformation function benefit in performance?
>
> The original intuition behind the proofreading layered stemmed from the architecture of the digit recognition component of our model. The digit classifier only takes individual sudoku cells into consideration, and consequently is unable to benefit from contextual information on the board. For example, if the digit classifier identifies two 5’s on the same row, it might intelligently infer that one of these can’t be a correct classification, and reconsider the lower-confidence prediction.
>
> The above intuition is not necessarily the sole reason for the improvement. As we can see in the results, the per-cell accuracy doesn’t strictly increase with the addition of the proofreader, and furthermore, the proofreader seems to benefit SATNet even in the non-visual setting, where all the input digit classifications are by definition correct. Thus, it is possible that the Proofreader also improves the capacity of the SATNet layer. A confirmation of this could potentially be the subject of future work.
>
> Lastly, the intuition behind the slightly noisy identity transform is that we wanted to take a fully trained model and improve upon it. Thus we initialize the Proofreader using the identity transform. The noise is there to give the optimizer a “kick”, making it less likely that it starts training at a flat region in loss space.
>
> > 5. In line 204, what is the precise definition of approxmax used in the experiment?
>
> We use the 2-norm as our approxmax function, and anecdotally found that our results aren’t significantly affected with other formulations. We mention this briefly on line 206.
>
> > 6. Why are experiments limited to Sudoku problem only? This limits the generalisation of the proposed formulation, at least empirically.
>
> We follow the work of [8], which calls attention to issues with SATNet's symbol-grounding ability specifically in the domain of MNIST/Sudoku - we here show how these problems may be fixed. We feel that our results with Visual Sudoku are indicative of the framework which we are presenting, and we hope to have convinced our readers of the same, even though certainly additional experimental setups in future work could be instructive. We discuss this further in our reply to question 1 above.
>
> > 7. In table 1, for “grounded” version of SATNet, if per-cell accuracy is 99%, why is total-board accuracy only 67%? Moreover, why is total-board accuracy around 65% for both grounded and ungrounded cases? Any intuition behind that? Is this an average result? If so, how many different Sudoku samples are considered in the experiment?
>
> We discuss this briefly in section 5.2, but agree that it would be better if we perhaps included a footnote clarifying it right after presenting the results in section 4. Essentially the discrepancy arises from the fact that there are on average 36.2 input cells per Sudoku board in our dataset. Even with the LeNet maximum of 99.2% digit recognition accuracy, the odds of identifying all the input cells correctly is only .992^(36.2) = 74.8%. This forms a rough upper bound for the performance of SATNet in this context (and we discuss this “roughness” in section 5.2).
>
> > 8. In Figure 4, what is the interpretation behind different colors in both figures?
>
> Thank you for bringing this up, we should definitely clarify this in the next revision. The colors come from the Matplotlib “Viridis” Color Bar, ranging from low values represented in purple to high values shown in yellow.

---

### Official Review · Reviewer_HyEa · 2021-07-16

**Rating:** 9
**Confidence:** 4

**Summary:**

The paper addresses how the symbol grounding problem arises in Visual Sudoku problem and proposes a self-supervised approach to grounding symbols in the SATNet architecture without relying on inadvertent label leakage. The leakage problem and its impact on SATNet for Visual Sudoku was known for previous work, so the paper’s key contributions revolve around a solution to this problem. In particular, it introduces a perceptual pre-training stage (clustering and distillation), a symbol grounding loss, and the performance boosting “proofreader” technique.

**Ethical Concerns:**

No concerns here.

**Limitations And Societal Impact:**

Choosing a clustering approach that is known to distinguish MNIST digits with 95% accuracy and then going on to apply it to MNIST digits feels like another form of inadvertent label leakage, though certainly not as egregious as the original form of leakage considered by the authors. Even telling the clustering algorithm to define 9 clusters (should it actually be 10 to represent the visual symbol “0”?) feels like a kind of leakage, even though it is much less significant. If we think of digit classification as multi-class classification with labels like “1”, “2”, “3”, clustering avoids leaking these labels. However, if we think of the *hard* parts of digit classification as being able to accurately judge sameDigit(a,b), then carefully chosen clustering approaches can leak lots of labels for this kind of digit classifier.

A revision of this paper should briefly discuss this potential problem of leaking pairwise categorization information. (This reviewer would love to see a paper on a taxonomy of things like label leakage as they pertain to the symbol grounding problem.)

I believe the authors are really getting at the heart of the symbol grounding problem with their work, but a small change in the experiment might allay concerns that clustering is implicitly leaking the hardest part of the task (identifying the useful categories): Try changing the number of clusters to something other than 9, say 100. We still need to solve the symbol grounding task to identify the 9 or 10 significant latent concepts, but this way we are pretty sure that the clustering is also including distracting concepts (e.g. different styles of “8”s) -- grounding would require recovering more than just a permutation.

**Main Review:**

The essence of the proposed approach is to identify the symbol grounding problem as resolving a mapping between latent perceptual categories and latent symbolic categories suggested by the symbolic labels in the training outputs. In the situation examined, this reduces to learning a permutation matrix (under the assumption that the two vocabularies have a 1:1 relation). Although this reduction is not particularly general, the solution offered (learning a linear alignment matrix) seems to be much more general by comparison. The combination of (1) self-supervised learning to anchor latent categories in the *input* images and (2) the symbol grounding loss to anchor the latent categories in the *output* symbols is novel, valuable, and thought provoking.

The proposal for the proofreader network seems unrelated to the core idea of the paper. If we think of the proofreader instead as a slight architectural improvement on LeNet, it’s not surprising that performance improves in both cases.

As an alternative to separate clustering and distillation, the authors might consider the setup used in VQVAEs and VQGANs (e.g. https://compvis.github.io/taming-transformers/). These networks learn to assign discrete codes to grids of tiles in a self-supervised manner. Replacing clustering and distillation in your work with VQGANs would not threaten your key exciting results. Using non-VQ VAEs or GANs could learn useful latent representation and leave their discretization to the proofreader network -- now possibly seen as a learned quantization/categorization layer.

Overall, while the paper is centered on a toy problem, it convincingly uses the focus on the toy problem to identify and advance solutions to some very wide-scale problems in AI.

**Time Spent Reviewing:**

2

---

> ### Author Response · Authors · 2021-08-10
> **Sincere Thanks! and Further Discussion**
>
> We’d like to thank you sincerely for your thoughtful review. We are so grateful that you find our work as exciting as we do! We would like to address some of the points brought up below.
>
> > 1. As an alternative to separate clustering and distillation, the authors might consider the setup used in VQVAEs and VQGANs (e.g. https://compvis.github.io/taming-transformers/).
>
> This is a good point, we could definitely explore alternatives to InfoGAN. The choice of clustering algorithm is abstracted away fairly cleanly from the rest of our proposed framework. Thus it should be straightforward to experiment with alternatives. Perhaps an ablation study with respect to these options might benefit a future revision of this paper, or a follow-up work.
>
> > 2. Choosing a clustering approach that is known to distinguish MNIST digits with 95% accuracy and then going on to apply it to MNIST digits feels like another form of inadvertent label leakage, though certainly not as egregious as the original form of leakage considered by the authors. Even telling the clustering algorithm to define 9 clusters (should it actually be 10 to represent the visual symbol “0”?) …
>
> Regarding the observations about our fixed cluster number, we agree that we are encoding some knowledge of the problem in our model. However, the only signal which we need to know with this approach is the number of label classes, which we consider “not that bad.”
>
> A more principled way to solve this problem, however, is discussed in section 6. In theory it should be possible to “over-cluster”, meaning that we could have multiple PTE dimensions map to the same LE dimension (invoking the notation from Figure 3). Basically it would require us to generalize the symbol grounding loss to implement a surjective mapping as opposed to a permutation. This way, it would not specifically matter how each digit is clustered, as long as the semantic dimension relevant to the given MAXSAT problem is captured in some subset of the clustering. For example, in the Visual Sudoku context, we might conceive of a stronger clustering algorithm which yields separate clusters for both digit number and digit thickness. This way the number “5” would be spread out across a collection of clusters representing both thick and thin handwritten 5's. The introduction of a surjective mapping in the self-grounded training would allow for this type of clustering to be supported. This innovation would then allow for an arbitrary number of clusters to be chosen.
>
> Regarding the representation of 0, we actually do not include this in the definition of our clusters. 0 is a placeholder which represents the output cells of a given Sudoku puzzle, and this is represented internally using a bitmask. We note that this generalizes to any MAXSAT problem, which will always require some form of masking between input and output variables.
>
> > 3. A revision of this paper should briefly discuss this potential problem of leaking pairwise categorization information. (This reviewer would love to see a paper on a taxonomy of things like label leakage as they pertain to the symbol grounding problem.)
>
> This is an excellent idea, we agree that this would indeed be a valuable resource. Another reviewer suggested looking at some RL-inspired methods for solving Symbol Grounding for the Visual Question Answering CLEVR dataset, which might benefit the breadth of such a taxonomy. We would need to spend further thought on how to formulate this, but we will aim to add the proposed section in our revision. We will follow up in this thread with updates while the discussion period is open.
>
> We also agree with the point on the pairwise categorization leakage, and will add a discussion about it to our revision.

---

> > ### Comment · Reviewer_HyEa · 2021-08-18
> > **Ideas for exploration**
> >
> > And thank you for the thoughtful rebuttal!
> >
> > The over-clustering / surjective mapping idea proposed sounds very appropriate. I see how this would allow you to not feed the model knowledge of the correct latent conceptual vocabulary while still having a good chance at getting almost the same results in the end. The key notion of their being 9 important classes would still be present in the labels, but it would no longer also be baked into the network architecture (in the hyperparameter defining the width of the matrix that currently encodes a permutation).
> >
> > Here are some variations on your experiment to consider in the future:
> >
> > - Use labels that encode irrelevant details (which might have distracting spurious correlations with perceptual details): Take the current labels and randomly multiply each one by +1 or -1. A model that correctly recovers the latent concepts and correctly reasons over them can't get 100% accuracy anymore, but it should approach a theoretical ideal 50% accuracy on each cell. Where the over-clustering idea made sure the model can ignore irrelevant input distinctions, this focuses on ignoring output distinctions when there's no true underlying explanation for them.
> >
> > - Use labels don't encode all of the relevant details:  Take the current labels and replace each by its parity (x%2). A model that correctly recovers the latent concepts and correctly reasons over them should still be able to approach 100% accuracy, but we hope it will do this by coming up with the correct latent solution before projecting it back into the ambiguous target vocabulary. I think humans could learn this "sudoku, except you binarize the results before turning them in" variation from examples.
> >
> > - Use labels combining these ideas: (x%2)*rand_sign to yield new labels in {-1,0,+1}.
> >
> > The idea behind these interventions is to have the equality relation on labels be a *noisy* view on the equality relation between the concepts we want the model to understand (now with false positives and false negatives). The way that labels for input cells are withheld in the submitted paper focused on missing data, but dealing with noisy data is a distinct issue. These could be combined (most excited), or treated separately (better to illustrate concepts). A dataset that provided (x%2)*sign style labels for all cells (even input cells) would address noise without any missing data.
> >
> > (Feel free to use these specific conceptual examples in your revision if you like.)

---

> > > ### Author Response · Authors · 2021-08-20
> > > **Interesting Ideas**
> > >
> > > That is a nicely structured way of formulating future experiments. These would indeed cover some interesting cases for noisy instance of a symbol grounding task. Thank you again for sharing your ideas, they will make for some interesting further discussion.

---

### Decision · Program_Chairs · 2021-09-27

**Decision:**

Accept (Spotlight)

**Comment:**

This is a wonderful paper for which reviewers showed significant support. In particular, reviewers view that the paper address a very important problem of symbol grounding and the proposed approach is novel, and effective. The authors are advised to take extensive feedback by reviewers into account for the final version.